# How a Family History of Allergic Diseases Influences Food Allergy in Children: The Japan Environment and Children’s Study

**DOI:** 10.3390/nu14204323

**Published:** 2022-10-15

**Authors:** Mayako Saito-Abe, Kiwako Yamamoto-Hanada, Kyongsun Pak, Shintaro Iwamoto, Miori Sato, Yumiko Miyaji, Hidetoshi Mezawa, Minaho Nishizato, Limin Yang, Natsuhiko Kumasaka, Tohru Kobayashi, Yukihiro Ohya

**Affiliations:** 1Medical Support Center for the Japan Environment and Children’s Study, National Center for Child Health and Development, Tokyo 157-8535, Japan; 2Department of Data Science, Clinical Research Center, National Center for Child Health and Development, Tokyo 157-8535, Japan

**Keywords:** atopic dermatitis, cohort study, food allergy, heredity, risk factor

## Abstract

The influence of family allergic history on food allergy in offspring in Japan is unknown. We analyzed data from a nationwide birth cohort study using logistic regression models to examine the associations of maternal, paternal, and both parental histories of allergic diseases (food allergy, atopic dermatitis, asthma, and rhinitis) with their child’s food allergy at 1.5 and 3 years of age. This analysis included 69,379 singleton full-term mothers and 37,179 fathers and their children. All parental histories of allergic diseases showed significant positive associations with their child’s food allergy. When both parents had a history of allergic diseases, the adjusted odds ratio (aOR) tended to be higher than when either parent had allergic diseases (*p* for trend < 0.0001). The highest aOR was detected when both parents had food allergy (2.60; 95% confidential interval, 1.58–4.27), and the aOR was 1.71 when either parent had food allergy (95% confidential interval, 1.54–1.91). The aORs were attenuated but still had significant positive associations after adjusting for the child’s atopic dermatitis, a risk factor for allergy development. In conclusion, all parental allergic diseases were significantly positively associated with their child’s food allergy. The effect of family history showed a stepwise increase in risk from either parent to both parents, and the highest risk of allergic disease was a parental history of food allergy.

## 1. Introduction

Food allergy commonly affects children, typically emerging in the last 10–15 years [1]. Various risk factors for food allergy onset have been identified [2,3,4,5,6]. Capturing risk factors for food allergy is vital to preventing and treating food allergy. We believe that a family history of allergic disease is a major risk factor for offspring food allergy onset. However, the results of studies investigating associations between family history of allergic diseases and offspring food allergy onset have been inconsistent.

While several past cohort studies detected significant associations of family history of allergic diseases with offspring food allergy [5,7,8,9,10], others detected no associations [4,11,12,13]. Several factors may have contributed to these inconsistent results. First, each study used different definitions of family history of allergy. Second, the adjusting factors for statistical analyses varied among the studies. It is well-known that early childhood percutaneous exposure to food proteins via eczema skin is the most critical risk factor for food allergy onset in recent years [14,15,16]. However, most studies that detected significant associations between family history of allergy with offspring food allergy onset did not adjust for the comorbidity of offspring atopic dermatitis in their statistical analyses [5,7,8,9,10]. Variations in adjusting factors between studies could thus produce inconsistent results of associations between family history of allergy and offspring food allergy.

To our knowledge, no large, nationwide prospective birth cohort studies have reported the associations of family history of allergy with offspring food allergy onset in Japan. This study aimed to investigate the associations of various patterns of parental allergic disease with their children’s food allergy, adjusting for the cofactors of children’s atopic dermatitis and using data from a national birth cohort study in Japan.

## 2. Methods

### 2.1. Study Design, Setting, and Participants

This was a nationwide, multicenter, prospective general birth cohort study: the Japan Environment and Children’s Study (JECS) [17,18,19,20,21,22,23]. A general population of 103,060 pregnancies/97,413 mothers and 49,279 fathers was enrolled in the JECS from January 2011 to March 2014. The 15 Regional Centers cover a wide geographical area of Japan (Hokkaido, Miyagi, Fukushima, Chiba, Kanagawa, Koshin, Toyama, Aichi, Kyoto, Osaka, Hyogo, Tottori, Kochi, Fukuoka, and South Kyushu/Okinawa). Eligibility criteria were as follows: (1) currently pregnant; (2) living in Study Area for the foreseeable future; (3) expected delivery between August 1, 2011, and mid-2014; and (4) ability to understand the Japanese language. In total, 104,062 fetal records were enrolled. The JECS registry is the University Hospital Medical Information Network (UMIN000030786). The JECS was conducted in accordance with the Ethical Guidelines for Medical and Health Research Involving Human Subjects in Japan. The JECS protocol was reviewed and approved by the Ministry of the Environment’s Institutional Review Board on Epidemiological Studies and the Ethics Committees of all participating institutions (No. 10091001). We obtained written informed consent from all participants.

### 2.2. Variables

Caregivers answered questionnaires regarding their child and family during pregnancy; as well as when their child was aged 6 months, 1, 1.5, 2, 2.5, and 3 years. We assessed the cumulative prevalence of a child’s food allergy as the outcome variable for this study. From the questionnaires, we defined the cumulative prevalence of food allergy in children using the caregiver’s reported physician diagnosis of their child’s food allergy at 1.5 and 3 years of age. Physician-diagnosed food allergy was defined as a positive answer from the caregiver to the question: “Has your child ever been diagnosed by a physician as having food allergy in the past 12 months?” We evaluated the maternal and paternal lifetime histories of allergic diseases as the exposure variables. The mothers and fathers separately reported their lifetime history of allergic diseases (atopic dermatitis, bronchial asthma, food allergy, and allergic rhinitis) via the questionnaire during pregnancy. We defined any allergic disease as any one of the allergic diseases listed above. We stratified the family allergic history according to the parents’ allergic histories: (i) neither parent, (ii) either parent, (iii) both parents. The fifteen JECS Study Areas conducted the questionnaire data input, and the study Program Office performed the data management. The present study used the fixed data set “jecs-ta-201901930-qsn”, released in October 2019.

### 2.3. Statistical Analysis

To examine the associations of parental history of allergic diseases with children’s food allergy outcomes, we used a logistic regression model for the child’s food allergy at 1.5 and 3 years of age, designating non-food allergy as a reference category. The results are presented as estimates of the unadjusted (univariate model) or adjusted (multivariable model) odds ratios (OR or aOR, respectively) and the associated 95% confidence intervals (CI). We estimated aORs using a multivariable logistic regression model to adjust for the following potential confounders based on previous studies. As the number of fathers was about half the number of mothers in the JECS, we created two groups for the statistical analyses: Group 1 included children with maternal information, regardless of paternal information, and Group 2 included children with both maternal and paternal information. We used the following models: Model 1, the environment in utero and birth (sex of the child, low birth weight (<2500 g), maternal education, and annual family income); Model 2, Model 1 plus the environment after birth (the season of birth, intake of breast milk at 1 month of age, the timing of starting solid foods, and the place of birth); and Model 3, Model 2 plus the onset of atopic dermatitis up to 6 months. We also implemented multiple imputations using a chained equation algorithm with 20 iterations (R package MICE, version 3.3.0.) to account for missing confounders and presented the results using multiple imputation data. We attributed a *p*-value for the estimates in the tables for reference. This study was exploratory, aiming to assess the magnitude of association between parental allergy status and their child’s food allergy. Therefore, we did not define the statistical significance of the *p*-value at the design stage. Additionally, we attributed a *p*-trend as a post-hoc analysis for the results of the logistic regression models in Group 2. All statistical analyses were performed using R statistical software, version 3.6.3 (The R Foundation, Vienna, Austria).

## 3. Results

### 3.1. Characteristics of the Study Population

A total of 69,379 singleton full-term mothers (Group 1) and 37,179 fathers (Group 2) without missing history of allergic disease data were included in the analysis (Figure 1). Table 1 shows the baseline characteristics of children. In children with maternal information, the cumulative prevalence of food allergy was 10.3% and 12.2% at 1.5 and 3 years old, respectively (Group 1). A similar cumulative food allergy prevalence was observed in children with paternal information (Group 2). The baseline characteristics were similar in the two groups. Table 2 shows the baseline characteristics of mothers and fathers, which were also similar in the two groups. The most common allergic disease among parents was allergic rhinitis, followed by atopic dermatitis.

### 3.2. The Associations of Maternal Allergic Diseases with Food Allergy in Children

Table 3 shows the associations of maternal allergic diseases with food allergy in children. All maternal allergic diseases showed significant positive associations with their child’s food allergy at 1.5 and 3 years of age. A similar tendency was observed in both groups. Maternal food allergy had the highest aOR with their child’s food allergy at 3 years old (Model 3; aOR, 1.91; 95% CI, 1.68–2.16 in Group 2). Maternal atopic dermatitis had the second-highest aOR with their child’s food allergy at 3 years old (Model 3; aOR, 1.51; 95% CI, 1.39–1.63 in Group 2). For most maternal allergic histories, there were minimal differences in the aORs after adjusting for the confounders in each model at both 1.5 and 3 years of age. However, for the maternal history of atopic dermatitis, the magnitude of the aOR was attenuated in Model 3 after adjusting for the child’s early-onset atopic dermatitis from Model 2 to Model 3 at both 1.5 and 3 years of age.

### 3.3. The Association of Paternal Allergic Diseases with Food Allergy in Children

Table 4 shows the association of paternal allergic diseases with food allergy in children. All paternal allergic diseases had significant positive associations with their child’s food allergy at 1.5 and 3 years of age, showing similar trends to those of maternal allergic diseases. Paternal food allergy (Model 3; aOR, 1.47; 95% CI, 1.25–1.72 in Group 2) and atopic dermatitis (Model 3; aOR, 1.46; 95% CI, 1.33–1.59 in Group 2) with their child’s food allergy at 3 years of age were the highest and second-highest aORs. All models showed the same tendency. Similar to the maternal results (Table 3), the magnitude of aOR for the paternal history of atopic dermatitis in Model 3 was attenuated after adjusting for the child’s early-onset atopic dermatitis from Model 2 to Model 3 at both 1.5 and 3 years of age.

### 3.4. The Associations of Either and Both Parental Allergic Diseases with Food Allergy in Children

Table 5 shows the associations of either and both parental allergic diseases with their child’s food allergy (Group 2). All allergic diseases of either and both parents showed significant positive associations with their child’s food allergy at 1.5 and 3 years of age. When both parents had histories of allergic diseases, the aOR with their child’s food allergy tended to be higher than when either parent had allergic diseases (*p* for trend < 0.0001). The highest aORs in Model 3 were both parents with food allergy (aOR, 2.60; 95% CI, 1.58–4.27) and either parent with food allergy (aOR, 1.71; 95% CI, 1.54–1.91). Both parents with atopic dermatitis showed the second-highest aOR in Model 3 (aOR, 2.08; 95% CI, 1.73–2.49), and that of either parent with atopic dermatitis was 1.51 (95% CI, 1.40–1.62). The aORs were most attenuated after adjusting for early-onset atopic dermatitis in children from Model 2 to Model 3 at both 1.5 and 3 years of age. Parental allergic rhinitis and asthma also showed significant positive associations with their child’s food allergy in all models; however, the aORs were lower than those of parental food allergy and atopic dermatitis.

## 4. Discussion

This study evaluated the associations of parental history of allergic diseases with children’s food allergy development at 1.5 and 3 years old using data collected from a nationwide birth cohort study in Japan. Histories of atopic dermatitis, food allergy, bronchial asthma, and allergic rhinitis in either or both parents were significantly positively associated with children’s food allergy. Parental food allergy and atopic dermatitis tended to show higher aORs for their child’s food allergy than parental asthma and allergic rhinitis.

We would like to highlight the several strengths of this study. First, we analyzed a large dataset collected from the general population across Japan, minimizing selection bias. Second, the results of our study using the JECS data provide a representative epidemiological signature of the Japanese population. Michikawa et al. [18] previously showed that the characteristics of JECS participants were similar to the vital population statistics collected from the Japanese government. Third, we adjusted for the confounding factor of the child’s history of atopic dermatitis which is the most significant risk factor for food allergy onset. Fourth, we divided the exposure variables into allergic disease categories and examined either parental or both parental histories for the analysis. This could be useful in clinical settings for an individual situation of family history in allergic diseases.

### 4.1. Differences in Parental Allergic Diseases

We demonstrated that both parents with food allergy showed the highest aOR for their child’s food allergy, and the second-highest aOR was for both parents with atopic dermatitis, even after adjusting for the child’s atopic dermatitis. Our results suggest that parental food allergy and atopic dermatitis, which are related to percutaneous allergen exposures, might have stronger influences than airway allergic diseases, such as asthma and allergic rhinitis. Similarly, Python et al. [24] reported that both parents with atopic dermatitis had the highest aOR, and both parents with food allergy had the second-highest aOR for their child’s food allergy; although, they did not adjust for the child’s atopic dermatitis. Although the underlying mechanisms are unclear, we speculate that the strong associations of parental atopic dermatitis and the child’s atopic dermatitis and food allergy are related to percutaneous allergen exposures. Concerning whether the influence of paternal or maternal history was stronger, no consistent results were obtained in the history of allergic diseases by type.

### 4.2. Influence of Infant Atopic Dermatitis

Several past cohort studies that adjusted for the child’s atopic dermatitis did not detect associations between family history of allergic diseases with their children’s food allergy. [4,11,12] By contrast, we showed that the parents’ history of allergic disease remained significantly associated with their child’s food allergy, even after adjusting for the child’s atopic dermatitis in this study. However, aORs in all three analyses of this study (for either and both parents), we found the most significant reductions in Model 3 after adjusting for atopic dermatitis onset within 6 months of age from Model 2 to Model 3. This result suggests that child’s early-onset atopic dermatitis might be a greater risk factor for developing food allergy in children than other environmental risk factors. These results coincide with past study findings [13,15,25].

### 4.3. Limitations

This study has several limitations. First, reporting biases inevitably arose in our study. The outcome assessments were not conducted by clinicians directly but were evaluated using questionnaires answered by the caregivers about the physician-diagnosed food allergy. We do not have a linkage system for medical record review in Japan by the Japanese government. Furthermore, oral food challenge tests were not used to define food allergy in this study; however, it would be unfeasible to conduct a food challenge test for nationwide birth cohorts outside of clinical settings. Second, parents only answered if they or their children had specific outcomes, so missing values for each outcome were uncertain. The inability to determine missing data/response rates for some outcomes means that bias could have influenced the study findings. Thus, the prevalence of diseases might have been underestimated. Third, we did not evaluate causal food allergens, which might have influenced our results.

As shown in this study, family history of allergic disease is one of the important risk factors for developing food allergy. However, the greatest risk factor is personal atopic dermatitis of children [4,12,13,26], and the only evidenced preventive intervention for food allergy is promoting the early and regular oral intake of common allergenic food [27,28,29,30]. We need to not only tell parents with allergies about the increased risk to their child, but also advise them on how to reduce it.

In summary, this study showed the positive associations of either and both parental allergic diseases with children’s food allergy at 1.5 and 3 years old in the Japanese population; the ORs tended to be higher when both parents had a history of allergic diseases. These ORs were maintained after adjusting for the child’s early-onset atopic dermatitis, which is the strongest risk factor for developing food allergy. Our results highlighted the heredity of allergic diseases in Japanese populations, and the effect of family history showed a stepwise increase in risk from one parent to both parents. We can share our findings with pregnant parents with allergic diseases in a clinical setting to discuss the future risk of their child’s food allergy onset after birth. Furthermore, we expect postnatal follow-up pathways will be established that are tailored to the risks of individual cases.

## Figures and Tables

**Figure 1 nutrients-14-04323-f001:**
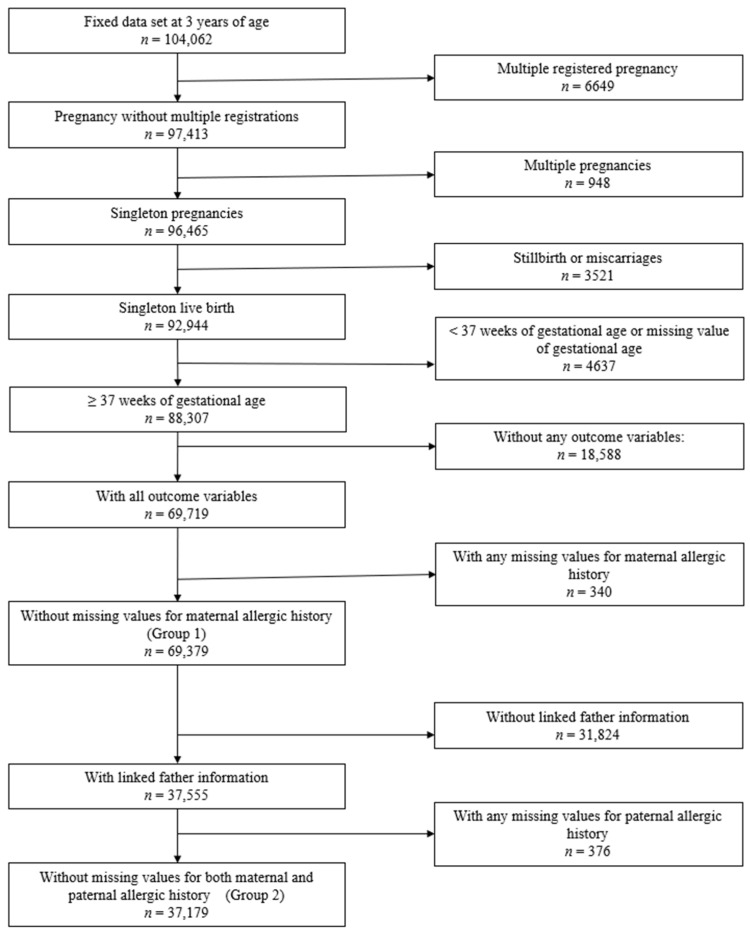
Flowchart of participant selection in this study.

**Table 1 nutrients-14-04323-t001:** Baseline characteristics of parents.

	Group 1	Group 2	
		Information from Mother	Information from Father
Age at delivery, years (mean ± SD)	31.5 ± 4.9	31.4 ± 4.8	-
BMI ^†^ at pregnancy, kg/m^2^ (mean ± SD)	21.1 ± 3.1	21.2 ± 3.2	-
Smoking during pregnancy	2741/69,379 (4.0%)	1305/37,179 (3.5%)	14,778/37,179 (39.7%)
Lifetime prevalence of atopic disease			
Atopic dermatitis	11,172/69,379 (16.1%)	6048/37,179 (16.3%)	4230/37,179 (10.7%)
Asthma	7306/69,379 (10.5%)	3936/37,179 (10.6%)	3979/37,179 (10.7%)
Allergic rhinitis	25,469/69,379 (36.7%)	13,861/37,179 (37.3%)	11,402/37,179 (30.7%)
Food allergy	3314/69,379 (4.8%)	1781/37,179 (4.8%)	1208/37,179 (3.2%)
Any allergy diseases	33,768/69,379 (48.7%)	18,305/37,179 (49.2%)	15,602/37,179 (42.0%)
Highest graduation			
Middle school	2445/68,860 (3.6%)	1138/36,952 (3.1%)	-
High School	50,210/68,860 (72.9%)	26,897/36,952 (72.3%)	-
University	16,195/68,860 (23.5%)	8917/36,952 (24.1%)	-
Place of recruitment			-
Hokkaido	5528/69,379 (8.0%)	2029/37,179 (5.5%)	-
Miygai/Fukushima	14,883/69,379 (21.5%)	9169/37,179 (24.7%)	-
Chiba/Kanagawa	8494/69,379 (12.2%)	4511/37,179 (12.1%)	-
Yamanashi/Koshu	4933/69,379 (7.1%)	3625/37,179 (9.8%)	-
Toyama	4031/69,379 (5.8%)	2495/37,179 (6.7%)	-
Aichi	3924/69,379 (5.7%)	1864/37,179 (5.0%)	-
Kyoto/Doshisha/Osaka/Hyogo	11,967/69,379 (17.2%)	5861/37,179 (15.8%)	-
Tottori	2159/69,379 (3.1%)	875/37,179 (2.4%)	-
Kochi	4516/69,379 (6.5%)	1646/37,179 (4.4%)	-
Kyushu	8406/69,379 (12.1%)	4858/37,179 (13.1%)	-
Okinawa	538/69,379 (0.8%)	246/37,179 (0.7%)	-
High annual family income (≥6 million yen)	18,081/64,760 (27.9%)	9822/34,983 (28.1%)	

† BMI: Body Mass Index. Group 1 included children with maternal information, regardless of paternal information; Group 2 included children with both maternal and paternal information. The information about highest graduation, place of recruitement, and annual family income was available only from mothers.

**Table 2 nutrients-14-04323-t002:** Baseline characteristics of children.

	Group 1	Group 2
Gender male	35,317/69,379 (50.9%)	18,882/37,179 (50.8%)
Pet	16,663/68,638 (24.3%)	9086/36,870 (24.6%)
Delivery method		
Vaginal delivery	57,149/69,206 (82.6%)	30,609/37,161 (82.4%)
Cesarean section	12,057/69,206 (17.4%)	6484/37,161 (17.4%)
Birth season		
Spring	16,139/69,379 (23.3%)	8437/37,179 (22.7%)
Summer	18,447/69,379 (26.6%)	9816/37,179 (26.4%)
Autumn	19,125/69,379 (27.6%)	10,471/37,179 (28.2%)
Winter	15,668/69,379 (22.6%)	8455/37,179 (22.7%)
Low birth weight (<2500 g)	3747/69,341 (5.4%)	1955/37,161 (5.3%)
Early introduction (<6 months) of solid foods	30,346/67,746 (44.8%)	16,449/36,336 (45.3%)
Breast feeding at 1 month of age	68,201/68,993 (98.9%)	36,602/37,020 (98.9%)
AD ^†^ onset until 6 months of age	9164/68,415 (13.4%)	4931/36,721 (13.4%)
Food allergy		
until 1.5 years	7119/69,379 (10.3%)	3791/37,179 (10.2%)
until 3 years	8467/69,379 (12.2%)	4511/37,179 (12.1%)

† AD: atopic dermatitis. Group 1 included children with maternal information, regardless of paternal information; Group 2 included children with both maternal and paternal information.

**Table 3 nutrients-14-04323-t003:** The association between history of maternal allergic diseases and children’s food allergy in Group1 and 2.

		Group 1					Group 2				
		Children with Food Allergy *n* (%)	Unadjusted	Model 1 ^†^	Model 2 ^‡^	Model 3 ^§^	Children with Food Allergy *n* (%)	Unadjusted	Model 1 ^†^	Model 2 ^‡^	Model 3 ^§^
			Crude OR (95% CI)	Adjusted OR (95% CI)	Adjusted OR (95% CI)	Adjusted OR (95% CI)		Crude OR (95% CI)	Adjusted OR (95% CI)	Adjusted OR (95% CI)	Adjusted OR (95% CI)
Until 1.5 years											
Atopic dermatitis	No	5479/58,207 (9.4%)	-	-	-	-	2881/31,131 (9.3%)	-	-	-	-
	Yes	1640/11,172 (14.7%)	1.66 (1.56–1.76)	1.64 (1.55–1.74)	1.62 (1.53–1.72)	1.40 (1.32–1.49)	910/6048 (15.0%)	1.74 (1.60–1.88)	1.72 (1.59–1.87)	1.69 (1.55–1.83)	1.44 (1.32–1.57)
Bronchial asthma	No	6152/62,073 (9.9%)	-	-	-	-	3258/33,243 (9.8%)	-	-	-	-
	Yes	967/7306 (13.2%)	1.39 (1.29–1.49)	1.41 (1.31–1.52)	1.42 (1.32–1.53)	1.30 (1.20–1.40)	533/3936 (13.5%)	1.44 (1.31–1.59)	1.47 (1.34–1.63)	1.49 (1.35–1.64)	1.35 (1.22–1.50)
Allergic rhinitis	No	4018/43,910 (9.2%)	-	-	-	-	2085/23,318 (8.9%)	-	-	-	-
	Yes	3101/25,469 (12.2%)	1.38 (1.31–1.45)	1.37 (1.30–1.44)	1.37 (1.30–1.44)	1.26 (1.20–1.33)	1706/13,861 (12.3%)	1.43 (1.34–1.53)	1.42 (1.33–1.52)	1.43 (1.33–1.53)	1.30 (1.21–1.40)
Food allergy	No	6557/66,065 (9.9%)	-	-	-	-	3468/35,398 (9.8%)	-	-	-	-
	Yes	562/3314 (17.0%)	1.85 (1.69–2.04)	1.85 (1.69–2.04)	1.85 (1.68–2.04)	1.73 (1.57–1.91)	323/1781 (18.1%)	2.04 (1.80–2.31)	2.04 (1.80–2.31)	2.03 (1.78–2.30)	1.91 (1.67–2.19)
Any atopic diseases	No	3005/35,611 (8.4%)	-	-	-	-	1525/18,874 (8.1%)	-	-	-	-
	Yes	4114/33,768 (12.2%)	1.51 (1.43–1.58)	1.50 (1.42–1.57)	1.49 (1.42–1.57)	1.34 (1.27–1.41)	2266/18,305 (12.4%)	1.61 (1.50–1.72)	1.60 (1.50–1.72)	1.60 (1.49–1.71)	1.42 (1.32–1.53)
Until 3 years											
Atopic dermatitis	No	6510/58,207 (11.2%)	-	-	-	-	3419/31,131 (11.0%)	-	-	-	-
	Yes	1957/11,172 (17.5%)	1.69 (1.60–1.78)	1.68 (1.59–1.77)	1.66 (1.57–1.75)	1.45 (1.36–1.53)	1092/6048 (18.1%)	1.79 (1.66–1.92)	1.77 (1.64–1.91)	1.74 (1.61–1.87)	1.51 (1.39–1.63)
Bronchial asthma	No	7316/62,073 (11.8%)	-	-	-	-	3876/33,243 (11.7%)	-	-	-	-
	Yes	1151/7306 (15.8%)	1.40 (1.31–1.50)	1.42 (1.33–1.52)	1.43 (1.34–1.53)	1.32 (1.23–1.41)	635/3936 (16.1%)	1.46 (1.33–1.60)	1.49 (1.36–1.63)	1.50 (1.37–1.64)	1.37 (1.25–1.51)
Allergic rhinitis	No	4796/43,910 (10.9%)	-	-	-	-	2499/23,318 (10.7%)	-	-	-	-
	Yes	3671/25,469 (14.4%)	1.37 (1.31–1.44)	1.37 (1.30–1.43)	1.37 (1.31–1.43)	1.27 (1.21–1.34)	2012/13,861 (14.5%)	1.41 (1.33–1.51)	1.41 (1.32–1.50)	1.41 (1.33–1.51)	1.30 (1.22–1.39)
Food allergy	No	7817/66,065 (11.8%)	-	-	-	-	4135/35,398 (11.7%)	-	-	-	-
	Yes	650/3314 (19.6%)	1.82 (1.66–1.99)	1.82 (1.66–1.99)	1.81 (1.66–1.98)	1.70 (1.55–1.87)	376/1781 (21.1%)	2.02 (1.80–2.28)	2.02 (1.80–2.28)	2.01 (1.78–2.26)	1.91 (1.68–2.16)
Any atopic diseases	No	3591/35,611 (10.1%)		-	-	-	1827/18,874 (9.7%)	-	-	-	-
	Yes	4876/33,768 (14.4%)	1.50 (1.44–1.58)	1.50 (1.43–1.57)	1.49 (1.43–1.57)	1.35 (1.29–1.42)	2684/18,305 (14.7%)	1.60 (1.50–1.71)	1.60 (1.50–1.70)	1.59 (1.50–1.70)	1.43 (1.34–1.53)

Group 1 included children with maternal information, regardless of paternal information; Group 2 included children with both maternal and paternal information. † Model 1; The environment in utero and birth included sex of the child, low birth weight, maternal education, and annual family income. ‡ Model 2; Model 1+ Living environment after birth included the season of birth, intake of breast milk at 1 month of age, timing of starting of solid foods, and the place of birth. § Model 3; Model 1 + 2, +the known risk factor for FA after birth, onset of AD up to 6 months of age.

**Table 4 nutrients-14-04323-t004:** The association between history of paternal allergic diseases and children’s food allergy in Group 2.

		Children with Food Allergy *n* (%)	Unadjusted	Model 1	Model 2	Model 3
			Crude OR (95% CI)	Adjusted OR (95% CI)	Adjusted OR (95% CI)	Adjusted OR (95% CI)
Until 1.5 years						
Atopic dermatitis	No	3146/32,949 (9.5%)	-	-	-	-
	Yes	645/4230 (15.2%)	1.70 (1.56–1.87)	1.70 (1.55–1.86)	1.68 (1.53–1.84)	1.43 (1.30–1.58)
Bronchial asthma	No	3266/33,200 (9.8%)	-	-	-	-
	Yes	525/3979 (13.2%)	1.39 (1.26–1.54)	1.39 (1.26–1.54)	1.40 (1.27–1.55)	1.29 (1.16–1.43)
Allergic rhinitis	No	2490/25,777 (9.7%)	-	-	-	-
	Yes	1301/11,402 (11.4%)	1.20 (1.12–1.29)	1.20 (1.11–1.28)	1.18 (1.10–1.27)	1.14 (1.05–1.22)
Food allergy	No	3610/35,971 (10.0%)	-	-	-	-
	Yes	181/1208 (15.0%)	1.58 (1.34–1.86)	1.59 (1.35–1.87)	1.57 (1.33–1.85)	1.44 (1.21–1.71)
Any atopic diseases	No	1959/21,577 (9.1%)	-	-	-	-
	Yes	1832/15,602 (11.7%)	1.33 (1.25–1.43)	1.32 (1.24–1.42)	1.31 (1.23–1.40)	1.21 (1.13–1.30)
Until 3 years						
Atopic dermatitis	No	3750/32,949 (11.4%)	-	-	-	-
	Yes	761/4230 (18.0%)	1.71 (1.57–1.86)	1.70 (1.56–1.85)	1.68 (1.54–1.83)	1.46 (1.33–1.59)
Bronchial asthma	No	3881/33,200 (11.7%)	-	-	-	-
	Yes	630/3979 (15.8%)	1.42 (1.30–1.56)	1.42 (1.30–1.56)	1.43 (1.30–1.57)	1.32 (1.20–1.46)
Allergic rhinitis	No	2966/25,777 (11.5%)	-	-	-	-
	Yes	1545/11,402 (13.6%)	1.21 (1.13–1.29)	1.20 (1.12–1.28)	1.19 (1.11–1.27)	1.15 (1.07–1.23)
Food allergy	No	4296/35,971 (11.9%)	-	-	-	-
	Yes	215/1208 (17.8%)	1.60 (1.37–1.86)	1.60 (1.38–1.87)	1.58 (1.36–1.84)	1.47 (1.25–1.72)
Any atopic diseases	No	2329/21,577 (10.8%)	-	-	-	-
	Yes	2182/15,602 (14.0%)	1.34 (1.26–1.43)	1.34 (1.26–1.42)	1.33 (1.25–1.41)	1.23 (1.16–1.32)

Group 2 included children with both maternal and paternal in-formation.

**Table 5 nutrients-14-04323-t005:** The association between history of parental allergic diseases and children’s food allergy in Group 2.

	Unadjusted				Model 1 ^†^				Model 2 ^‡^				Model 3 ^§^			
	Neither Parent (Reference)	Either Parent aOR (95% CI)	Both Parents aOR (95% CI)	*p* Trend	Neither Parent (Reference)	Either Parent aOR (95% CI)	Both Parents aOR (95% CI)	*p* Trend	Neither Parent (Reference)	Either Parent aOR (95% CI)	Both Parents aOR (95% CI)	*p* Trend	Neither Parent (Reference)	Either Parent aOR (95% CI)	Both Parents aOR (95% CI)	*p* Trend
Until 1.5 years																
Atopic dermatitis	-	1.75 (1.63–1.88)	2.68 (2.24–3.21)	<0.0001	-	1.74 (1.61–1.87)	2.67 (2.23–3.20)	<0.0001	-	1.71 (1.59–1.84)	2.60 (2.17–3.13)	<0.0001	-	1.46 (1.36–1.58)	1.92 (1.58–2.33)	*p* < 0.0001
Bronchial asthma	-	1.44 (1.33–1.56)	1.77 (1.38–2.27)	<0.0001	-	1.46 (1.34–1.58)	1.80 (1.40–2.31)	<0.0001	-	1.47 (1.36–1.59)	1.81 (1.41–2.33)	<0.0001	-	1.37 (1.26–1.49)	1.38 (1.05–1.80)	0.0192
Allergic rhinitis	-	1.35 (1.25–1.45)	1.67 (1.51–1.84)	<0.0001	-	1.34 (1.24–1.44)	1.65 (1.49–1.83)	<0.0001	-	1.34 (1.24–1.44)	1.64 (1.48–1.82)	<0.0001	-	1.27 (1.18–1.37)	1.43 (1.28–1.59)	<0.0001
Food allergy	-	1.83 (1.65–2.04)	2.98 (1.84–4.83)	<0.0001	-	1.84 (1.65–2.04)	3.00 (1.85–4.86)	<0.0001	-	1.82 (1.64–2.03)	2.96 (1.82–4.83)	<0.0001	-	1.71 (1.53–1.91)	2.49 (1.47–4.21)	0.0007
Any atopic diseases	-	1.45 (1.33–1.58)	2.07 (1.89–2.28)	<0.0001	-	1.45 (1.33–1.58)	2.06 (1.87–2.26)	<0.0001	-	1.44 (1.32–1.57)	2.03 (1.85–2.24)	<0.0001	-	1.34 (1.23–1.47)	1.69 (1.53–1.86)	<0.0001
Until 3 years																
Atopic dermatitis	-	1.77 (1.66–1.90)	2.81 (2.38–3.33)	<0.0001	-	1.76 (1.64–1.89)	2.81 (2.37–3.32)	<0.0001	-	1.73 (1.62–1.86)	2.74 (2.31–3.25)	<0.0001	-	1.51 (1.40–1.62)	2.08 (1.73–2.49)	<0.0001
Bronchial asthma	-	1.46 (1.35–1.57)	1.87 (1.49–2.35)	<0.0001	-	1.47 (1.37–1.58)	1.90 (1.51–2.39)	<0.0001	-	1.48 (1.38–1.60)	1.91 (1.51–2.41)	<0.0001	-	1.39 (1.29–1.50)	1.49 (1.16–1.90)	0.0017
Allergic rhinitis	-	1.34 (1.25–1.44)	1.65 (1.50–1.81)	<0.0001	-	1.33 (1.25–1.43)	1.64 (1.49–1.80)	<0.0001	-	1.34 (1.25–1.43)	1.63 (1.48–1.80)	<0.0001	-	1.27 (1.19–1.37)	1.44 (1.30–1.59)	<0.0001
Food allergy	-	1.83 (1.65–2.02)	3.08 (1.95–4.85)	<0.0001	-	1.83 (1.66–2.02)	3.08 (1.95–4.87)	<0.0001	-	1.81 (1.64–2.01)	3.04 (1.91–4.82)	<0.0001	-	1.71 (1.54–1.91)	2.60 (1.58–4.27)	0.0002
Any atopic diseases	-	1.44 (1.33–1.56)	2.08 (1.91–2.28)	<0.0001	-	1.44 (1.33–1.56)	2.07 (1.90–2.26)	<0.0001	-	1.44 (1.33–1.56)	2.06 (1.88–2.25)	<0.0001	-	1.35 (1.24–1.46)	1.73 (1.58–1.90)	<0.0001

Group 2 included children with both maternal and paternal information. † Model 1; The environment in utero and birth included sex of the child, low birth weight, maternal education, and annual family income. ‡ Model 2; Model 1+ Living environment after birth included the season of birth, intake of breast milk at 1 month of age, timing of starting of solid foods, and the place of birth. § Model 3; Model 1 + 2, +the known risk factor for FA after birth, onset of AD up to 6 months of age.

## Data Availability

Data sharing not applicable.

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
