# Peer review of "How a Family History of Allergic Diseases Influences Food Allergy in Children: The Japan Environment and Children’s Study"

_nutrients, 2022, doi:10.3390/nu14204323_

Round 1
Reviewer 1 Report
The authors have examined whether family history of atopic disease increase the likelihood of food allergy in their children. By examining a national prospective birth cohort based in Japan, the presence of atopic disease, including food allergy was defined by questionnaires administered to caregivers, with food allergy, defined as a caregivers reported physician diagnosis of food allergy up to the age of 3 years. They describe the cumulative presence of food allergy as being around 10.3% and 4.2% and 1.5 and 3 years, respectively, when information was provided by information provided by mothers and fathers. While the presence of atopic disease overall in either parent increased likelihood of reported food allergy and their offspring, food allergy and eczema in both mothers and fathers was associated with the highest risk ratio of food allergy in their offspring compared to inhalant allergy alone. Furthermore, the presence of atopic disease in both parents conveyed a greater risk of food allergy as an outcome than the presence of atopic disease in one parent alone.
Comments
This is a study which builds on other studies examining the relationship between family history of atopic disease and food allergy outcome. The authors help distinguish that family history of allergic disease increases the risk of food allergy in children, independent of whether the children have eczema or not. Although the presence of eczema is a significant an independent risk factor. Similar published studies are summarised below, have shown conflicting evidence of whether family history is a major risk factor, with some suggesting that in the absence of eczema, family history plays no role. On balance, however, most published studies show similar outcomes to this Japanese study. The major caveat as discussed by the author's is that food allergy was based on history alone and not testing or challenge. The author suggests that family history is a risk factor can be discussed with pregnant mothers stating that their own history can increase the risk of food allergy in their children. I would suggest however that the only intervention we really have been shown to reduce the risk of food allergy is early and regular consumption of common allergenic food. Enhancing that discussion with parents they go some way to reducing the risk of food allergy in their offspring rather than simply telling the parents that child is at higher risk than average. I would add this to the discussion.
1: Koplin JJ, Allen KJ, Gurrin LC, Peters RL, Lowe AJ, Tang ML, Dharmage SC;
HealthNuts Study Team. The impact of family history of allergy on risk of food
allergy: a population-based study of infants. Int J Environ Res Public Health.
2013 Oct 25;10(11):5364-77. doi: 10.3390/ijerph10115364. PMID: 24284354; PMCID:
PMC3863850.
2: Gupta RS, Walkner MM, Greenhawt M, Lau CH, Caruso D, Wang X, Pongracic JA,
Smith B. Food Allergy Sensitization and Presentation in Siblings of Food
Allergic Children. J Allergy Clin Immunol Pract. 2016 Sep-Oct;4(5):956-62. doi:
10.1016/j.jaip.2016.04.009. Epub 2016 Jul 12. PMID: 27421900; PMCID: PMC5010481.
3: Kalach N, Bellaïche M, Elias-Billon I, Dupont C. Family history of atopy in
infants with cow's milk protein allergy: A French population-based study. Arch
Pediatr. 2019 May;26(4):226-231. doi: 10.1016/j.arcped.2019.02.014. Epub 2019
Mar 15. PMID: 30885604.
4: Sasaki M, Peters RL, Koplin JJ, Field MJ, McWilliam V, Sawyer SM, Vuillermin
PJ, Pezic A, Gurrin LC, Douglass JA, Tang MLK, Dharmage SC, Allen KJ. Risk
Factors for Food Allergy in Early Adolescence: The SchoolNuts Study. J Allergy
Clin Immunol Pract. 2018 Mar-Apr;6(2):496-505. doi: 10.1016/j.jaip.2017.12.011.
Epub 2018 Feb 1. PMID: 29397374.
5: Keet C, Pistiner M, Plesa M, Szelag D, Shreffler W, Wood R, Dunlop J, Peng R,
Dantzer J, Togias A. Age and eczema severity, but not family history, are major
risk factors for peanut allergy in infancy. J Allergy Clin Immunol. 2021
Mar;147(3):984-991.e5. doi: 10.1016/j.jaci.2020.11.033. Epub 2021 Jan 19. PMID:
33483153; PMCID: PMC8462937.
6: Tan TH, Ellis JA, Saffery R, Allen KJ. The role of genetics and environment
in the rise of childhood food allergy. Clin Exp Allergy. 2012 Jan;42(1):20-9.
doi: 10.1111/j.1365-2222.2011.03823.x. Epub 2011 Jul 19. PMID: 21771119.
Author Response
Comments
This is a study which builds on other studies examining the relationship between family history of atopic disease and food allergy outcome. The authors help distinguish that family history of allergic disease increases the risk of food allergy in children, independent of whether the children have eczema or not. Although the presence of eczema is a significant an independent risk factor. Similar published studies are summarised below, have shown conflicting evidence of whether family history is a major risk factor, with some suggesting that in the absence of eczema, family history plays no role. On balance, however, most published studies show similar outcomes to this Japanese study.
Response: Thank you for pointing out the previous studies, including ones we didn't cite in the manuscript. When we summarized the past literature in the introduction, we focused on prospective cohort studies in consideration of the level of evidence. We have added the review article you pointed out to the reference.
- Tan, T.H.; Ellis, J.A.; Saffery, R.; Allen, K.J. The role of genetics and environment in the rise of childhood food allergy. Clin Exp Allergy 2012, 42, 20-29, doi:10.1111/j.1365-2222.2011.03823.x.
The major caveat as discussed by the author's is that food allergy was based on history alone and not testing or challenge. The author suggests that family history is a risk factor can be discussed with pregnant mothers stating that their own history can increase the risk of food allergy in their children. I would suggest however that the only intervention we really have been shown to reduce the risk of food allergy is early and regular consumption of common allergenic food. Enhancing that discussion with parents they go some way to reducing the risk of food allergy in their offspring rather than simply telling the parents that child is at higher risk than average. I would add this to the discussion.
Response: Thank you for your important suggestions. We have added the comment about what caregivers should do to reduce the risk of the child’s food allergy.
Page 10, Line 244-249
As shown in this study, family history of allergic disease is one of the important risk factors for developing food allergy. However, the greatest risk factor is personal atopic dermatitis of children[4,12-13,26], and the only evidenced preventive intervention for food allergy is promoting the early and regular oral intake of common allergenic food[27-30]. We need to not only tell parents with allergies about the increased risk to the child, but also advise them on what to do to reduce it.
Along with this addition, the following reference was added.
- Al-Hammadi, S.; Zoubeidi, T.; Al-Maskari, F. Predictors of childhood food allergy: significance and implications. Asian Pac J Allergy Immunol 2011, 29, 313-317.
- Bellach, J.; Schwarz, V.; Ahrens, B.; Trendelenburg, V.; Aksünger, Ö.; Kalb, B.; Niggemann, B.; Keil, T.; Beyer, K. Randomized placebo-controlled trial of hen's egg consumption for primary prevention in infants. J Allergy Clin Immunol 2017, 139, 1591-1599.e1592, doi:10.1016/j.jaci.2016.06.045.
- Du Toit, G.; Roberts, G.; Sayre, P.H.; Bahnson, H.T.; Radulovic, S.; Santos, A.F.; Brough, H.A.; Phippard, D.; Basting, M.; Feeney, M., et al. Randomized trial of peanut consumption in infants at risk for peanut allergy. N Engl J Med 2015, 372, 803-813, doi:10.1056/NEJMoa1414850.
- Natsume, O.; Kabashima, S.; Nakazato, J.; Yamamoto-Hanada, K.; Narita, M.; Kondo, M.; Saito, M.; Kishino, A.; Takimoto, T.; Inoue, E., et al. Two-step egg introduction for prevention of egg allergy in high-risk infants with eczema (PETIT): a randomised, double-blind, placebo-controlled trial. Lancet 2017, 389, 276-286, doi:10.1016/s0140-6736(16)31418-0.
- Palmer, D.J.; Sullivan, T.R.; Gold, M.S.; Prescott, S.L.; Makrides, M. Randomized controlled trial of early regular egg intake to prevent egg allergy. J Allergy Clin Immunol 2017, 139, 1600-1607.e1602, doi:10.1016/j.jaci.2016.06.052.
1: Koplin JJ, Allen KJ, Gurrin LC, Peters RL, Lowe AJ, Tang ML, Dharmage SC;
HealthNuts Study Team. The impact of family history of allergy on risk of food
allergy: a population-based study of infants. Int J Environ Res Public Health.
2013 Oct 25;10(11):5364-77. doi: 10.3390/ijerph10115364. PMID: 24284354; PMCID:
PMC3863850.
2: Gupta RS, Walkner MM, Greenhawt M, Lau CH, Caruso D, Wang X, Pongracic JA,
Smith B. Food Allergy Sensitization and Presentation in Siblings of Food
Allergic Children. J Allergy Clin Immunol Pract. 2016 Sep-Oct;4(5):956-62. doi:
10.1016/j.jaip.2016.04.009. Epub 2016 Jul 12. PMID: 27421900; PMCID: PMC5010481.
3: Kalach N, Bellaïche M, Elias-Billon I, Dupont C. Family history of atopy in
infants with cow's milk protein allergy: A French population-based study. Arch
Pediatr. 2019 May;26(4):226-231. doi: 10.1016/j.arcped.2019.02.014. Epub 2019
Mar 15. PMID: 30885604.
4: Sasaki M, Peters RL, Koplin JJ, Field MJ, McWilliam V, Sawyer SM, Vuillermin
PJ, Pezic A, Gurrin LC, Douglass JA, Tang MLK, Dharmage SC, Allen KJ. Risk
Factors for Food Allergy in Early Adolescence: The SchoolNuts Study. J Allergy
Clin Immunol Pract. 2018 Mar-Apr;6(2):496-505. doi: 10.1016/j.jaip.2017.12.011.
Epub 2018 Feb 1. PMID: 29397374.
5: Keet C, Pistiner M, Plesa M, Szelag D, Shreffler W, Wood R, Dunlop J, Peng R,
Dantzer J, Togias A. Age and eczema severity, but not family history, are major
risk factors for peanut allergy in infancy. J Allergy Clin Immunol. 2021
Mar;147(3):984-991.e5. doi: 10.1016/j.jaci.2020.11.033. Epub 2021 Jan 19. PMID:
33483153; PMCID: PMC8462937.
6: Tan TH, Ellis JA, Saffery R, Allen KJ. The role of genetics and environment
in the rise of childhood food allergy. Clin Exp Allergy. 2012 Jan;42(1):20-9.
doi: 10.1111/j.1365-2222.2011.03823.x. Epub 2011 Jul 19. PMID: 21771119.

Reviewer 2 Report
General comments
Out of a nationwide cohort study initiated 2014 (The Japan Environment and Children´s Study) questionnaires were provided and 69,379 singleton full-term mothers (Group 1) and 37,179 fathers (Group 2) gave history of allergic disease data of the parents and their child at different time points of age (6 months, 1, 1.5, 2, 2.5, and 3 years). Cumulative prevalence of a child´s food allergy represented the primary outcome variable for the present study. Prevalence of atopic dermatitis was equally assessed in the child in order to adjust for this outcome. In particular, the caregiver´s reported physician diagnosis of the child´s food allergy was asked for. In addition, mothers and fathers separately reported their lifetime history of allergic diseases (atopic dermatitis, bronchial asthma, food allergy, allergic rhinitis) via the questionnaire during pregnancy.
While the overall question of this study is highly relevant, the methods used to acquire these data if very questionable. As the authors altogether highlight in their study limitation part, there are many obstacles that hamper the proper interpretation of this study.
“First, reporting biases inevitably arose in our study. The outcome assessments were not conducted by clinicians but were evaluated by us in questionnaires answered by the caregivers. We do not have a linkage system for medical record review in Japan by the Japanese government.”
Many studies conducted in the last decades covering the question of food allergy prevalence in different areas of the continent revealed strongly overestimated values in questionnaire based-assessments. Thus, only validation processes with a) physician-based interviews, b) use of biomarkers such as Skin Prick Test or IgE-evaluation, and c) food challenges would be able to correct for these biases.
“Furthermore, oral food challenge tests were not used to define food allergy in this study; however, it would be unfeasible to conduct a food challenge test for birth cohorts outside of clinical settings.”
A way to overcome testing all food allergy suspected children in the study would be to take a representative regional center in which a proper food allergy evaluation would have been performed. Based on these (representative) data the authors could somehow extrapolate for the other study areas. However, such an approach must be thoroughly accompanied by statistical processes.
I wonder why the authors did not approach reknown allergology experts in the field in Japan to discuss the issues and how to overcome these points.
As recently outlined by Hossny et al in the World Allergy Organization Journal (WAO 2019) “a global increase in prevalence of asthma, allergic rhinitis, and atopic dermatitis, followed by a rapid rise in FA has been termed the "second wave of the allergy epidemic". Although we know of the global trend, the patterns of FA are highly variable in different parts of the world. These differences can be attributed partly to the many difficulties encountered in reaching an accurate estimation of prevalence. Self-report leads to an overestimating of prevalence by threefold to fourfold. Furthermore, widely varying study methodologies, the lack in the use of objective methods, and differences in baseline populations limit robust comparisons between populations.”
In particular in Asia data on food allergy prevalence is quite variable (Hossny et al, WAO 2019), and thus only proper methods will allow to define correct prevalence values.
The cumulative food allergy prevalence estimates of 10.3% and 12.2% at 1.5 and 3 years of age might be highly overestimated (as compared to data form US, Europe) or possibly in line with increasing prevalences of food allergy in Asia (as shown for Korea). However, without correct validation of the data we will not be able to substantiate this enigma.
Author Response
General comments
Out of a nationwide cohort study initiated 2014 (The Japan Environment and Children´s Study) questionnaires were provided and 69,379 singleton full-term mothers (Group 1) and 37,179 fathers (Group 2) gave history of allergic disease data of the parents and their child at different time points of age (6 months, 1, 1.5, 2, 2.5, and 3 years). Cumulative prevalence of a child´s food allergy represented the primary outcome variable for the present study. Prevalence of atopic dermatitis was equally assessed in the child in order to adjust for this outcome. In particular, the caregiver´s reported physician diagnosis of the child´s food allergy was asked for. In addition, mothers and fathers separately reported their lifetime history of allergic diseases (atopic dermatitis, bronchial asthma, food allergy, allergic rhinitis) via the questionnaire during pregnancy.
While the overall question of this study is highly relevant, the methods used to acquire these data if very questionable. As the authors altogether highlight in their study limitation part, there are many obstacles that hamper the proper interpretation of this study.
Response: Thank you for your valuable comments.
“First, reporting biases inevitably arose in our study. The outcome assessments were not conducted by clinicians but were evaluated by us in questionnaires answered by the caregivers. We do not have a linkage system for medical record review in Japan by the Japanese government.”
Many studies conducted in the last decades covering the question of food allergy prevalence in different areas of the continent revealed strongly overestimated values in questionnaire based-assessments. Thus, only validation processes with a) physician-based interviews, b) use of biomarkers such as Skin Prick Test or IgE-evaluation, and c) food challenges would be able to correct for these biases.
Response: Thank you for pointing this out. The sentence “The outcome assessments were not conducted by clinicians but were evaluated by us in questionnaires answered by the caregivers.” contained misleading expressions. Although this is a questionnaire survey, it is not a simple parental decision to report food allergy, but to collect the diagnosis made by a physician. To emphasize the point, we revised this sentence and added the explanation about specific content of the questionnaire as below.
Page 10, line 233-235
The outcome assessments were not conducted by clinicians directly but were evaluated by us in questionnaires answered by the caregivers about the physician-diagnosed food allergy.
Page 2, line 77-79
Physician-diagnosed food allergy was defined as a positive answer from the caregiver to the question: “Has your child ever been diagnosed by a physician as having food allergy in the past 12 months?”
“Furthermore, oral food challenge tests were not used to define food allergy in this study; however, it would be unfeasible to conduct a food challenge test for birth cohorts outside of clinical settings.”
A way to overcome testing all food allergy suspected children in the study would be to take a representative regional center in which a proper food allergy evaluation would have been performed. Based on these (representative) data the authors could somehow extrapolate for the other study areas. However, such an approach must be thoroughly accompanied by statistical processes.
I wonder why the authors did not approach reknown allergology experts in the field in Japan to discuss the issues and how to overcome these points.
Response: We well understand that the gold standard for the definitive diagnosis of food allergy is conducting food challenge test, and that it is also very important to confirm the presence of sensitization of the causal foods. However, it is not financially and ethically feasible to conduct food allergy evaluation at regional centers for all participants who report FA in this nation-wide population-based study (lack of feasibility for 100,100 populations). In addition, if we examined our hypothesis in the hospital-based surveys, which can be a retrospective case-control study, would bias the sample (population with very high pre-test probability) and there would be no control group, making it impossible to generalize the results. In fact, various studies assessing food allergy by questionnaire survey are reported.
Actually, we also conducted an on-site medical survey including blood sampling of 5,000 people in the JECS, in which we confirmed the presence of sensitization by specific IgE measurement from two years of age. We would like to report the study using these results in the future, but sample size would be small. The purpose of the current study was to investigate the association between family history and child’s food allergy early childhood, based on the big data of 100,000 nation-wide people.
As recently outlined by Hossny et al in the World Allergy Organization Journal (WAO 2019) “a global increase in prevalence of asthma, allergic rhinitis, and atopic dermatitis, followed by a rapid rise in FA has been termed the "second wave of the allergy epidemic". Although we know of the global trend, the patterns of FA are highly variable in different parts of the world. These differences can be attributed partly to the many difficulties encountered in reaching an accurate estimation of prevalence. Self-report leads to an overestimating of prevalence by threefold to fourfold. Furthermore, widely varying study methodologies, the lack in the use of objective methods, and differences in baseline populations limit robust comparisons between populations.”
In particular in Asia data on food allergy prevalence is quite variable (Hossny et al, WAO 2019), and thus only proper methods will allow to define correct prevalence values.
Response: We also understand that there is often a discrepancy between parent-reported illness and the physician’s diagnosis. As mentioned above, one thing we would like to emphasize is that although this is a questionnaire survey, it is not a simple parental decision to report food allergy, but to collect the diagnosis made by a physician.
The cumulative food allergy prevalence estimates of 10.3% and 12.2% at 1.5 and 3 years of age might be highly overestimated (as compared to data form US, Europe) or possibly in line with increasing prevalences of food allergy in Asia (as shown for Korea). However, without correct validation of the data we will not be able to substantiate this enigma.
Response:
We are the first to conduct a nationwide survey of food allergy prevalence in Japan, which certainly makes comparisons difficult. In a cross-sectional survey (n=3,345) of health check-ups at age 3 in Tokyo, it was reported that 17.8% had food allergy symptoms by age 3 and 14.9% were diagnosed by a doctor (Metropolitan survey of 3-year-old children on allergic diseases.2019. Japanese report by the Administration).
The other study conducted in Sagamihara city reported that the rate of food elimination due to food allergy at 12 m and 3 y was 12.8%, and 5.1%.
(M. Ebisawa, C. Sugizaki Prevalence of pediatric allergic diseases in the first 5 years of life J Allergy Clin Immunol, 121 (2008), p. S237).
Given these results, the cumulative food allergy prevalence up to the age of 3 years reported here by us are largely consistent. It is unavoidable that these results are also questionnaire-based, as they are prevalence studies in the general population.
Although it is difficult to resolve the problems with diagnostic accuracy that you have pointed out for us, prospective birth cohort study has the highest level of evidence for observational study and help to understand the state of the disease in the general population.

Reviewer 3 Report
I read with interest this manuscript aiming at assessing the influence of parental allergy history on food allergy in their offspring. This is a multi-centered, questionnaire based, prospective general birth cohort study with a very large data set.
Length of the introduction is good respect to the length of the manuscript. It reviews the background and states the objectives adequately. However, in the first sentence of the introduction (lines 32-33) a misinterpretation of reference number 1 is reported. The sense of that article is that food allergy has increased in the last 10-15 years (referring to 2011).
The results are interesting because they confirm for the Japanese population what is more or less already known about the effect of the positive history of allergies on parents and their offspring. More importantly, they lay the groundwork for a personalization of follow-up paths according to the extent of the risk of developing allergies through the primary and secondary prevention of these diseases.
In addition to the results already obtained, it would be interesting to know if maternal history of allergy has a greater effect than paternal one. Could you verify this?
Strengths and limitations are well acknowledged.
The references are fairly recent.
Tables are necessary and self-explaining especially with their clear captions.
Will questionnaires be available for readers?
Author Response
I read with interest this manuscript aiming at assessing the influence of parental allergy history on food allergy in their offspring. This is a multi-centered, questionnaire based, prospective general birth cohort study with a very large data set.
Length of the introduction is good respect to the length of the manuscript. It reviews the background and states the objectives adequately. However, in the first sentence of the introduction (lines 32-33) a misinterpretation of reference number 1 is reported. The sense of that article is that food allergy has increased in the last 10-15 years (referring to 2011).
Response: Thank you for pointing out the misinterpretation of reference number 1. We have corrected the sentence as below.
Page 1, Line 32-33.
Food allergy commonly affects children, typically emerging in the last 10–15 years of life.
The results are interesting because they confirm for the Japanese population what is more or less already known about the effect of the positive history of allergies on parents and their offspring. More importantly, they lay the groundwork for a personalization of follow-up paths according to the extent of the risk of developing allergies through the primary and secondary prevention of these diseases.
Response: We totally agree that establishing the follow-up paths to prevent food allergy in high- risk populations is most important goal. We have added the point in the conclusion as below.
Page 11, Line 258-259.
Furthermore, we expect postnatal follow-up pathways will be established that are tailored to the risks of individual cases.
In addition to the results already obtained, it would be interesting to know if maternal history of allergy has a greater effect than paternal one. Could you verify this?
Response: Thank you for pointing out the important issue. Although not included in the present manuscript, we had also analysed the influence of family history in mothers only and in fathers only. Concerning whether the influence of paternal or maternal history was stronger, no consistent results were obtained in the history of allergic diseases by type. Furthermore, the difference in the magnitude of the odds between mother and father was not large enough to be of interest. Therefore, we thought it would be easier to get the message across if we summarised the results by whether it was one or both parents in this study.
We added this point as below:
Page 10, line 217-219
Concerning whether the influence of paternal or maternal history was stronger, no consistent results were obtained in the history of allergic diseases by type (data not shown).
Strengths and limitations are well acknowledged.
The references are fairly recent.
Tables are necessary and self-explaining especially with their clear captions.
Will questionnaires be available for readers?
Response: The JECS study is now preparing to release the questionnaires we used to the public in the future, but this has not yet been achieved at this moment.

Round 2
Reviewer 2 Report
Out of a nationwide cohort study initiated 2014 (The Japan Environment and Children´s Study) questionnaires were provided and 69,379 singleton full-term mothers (Group 1) and 37,179 fathers (Group 2) gave history of allergic disease data of the parents and their child at different time points of age (6 months, 1, 1.5, 2, 2.5, and 3 years). Cumulative prevalence of a child´s food allergy represented the primary outcome variable for the present study. Prevalence of atopic dermatitis was equally assessed in the child in order to adjust for this outcome. In particular, the caregiver´s reported physician diagnosis of the child´s food allergy was asked for. In addition, mothers and fathers separately reported their lifetime history of allergic diseases (atopic dermatitis, bronchial asthma, food allergy, allergic rhinitis) via the questionnaire during pregnancy.
While the overall question of this study is highly relevant, the methods used to acquire these data if very questionable. As the authors altogether highlight in their study limitation part, there are many obstacles that hamper the proper interpretation of this study.
“First, reporting biases inevitably arose in our study. The outcome assessments were not conducted by clinicians but were evaluated by us in questionnaires answered by the caregivers. We do not have a linkage system for medical record review in Japan by the Japanese government.”
Many studies conducted in the last decades covering the question of food allergy prevalence in different areas of the continent revealed strongly overestimated values in questionnaire based-assessments. Thus, only validation processes with a) physician-based interviews, b) use of biomarkers such as Skin Prick Test or IgE-evaluation, and c) food challenges would be able to correct for these biases.
“Furthermore, oral food challenge tests were not used to define food allergy in this study; however, it would be unfeasible to conduct a food challenge test for birth cohorts outside of clinical settings.”
A way to overcome testing all food allergy suspected children in the study would be to take a representative regional center in which a proper food allergy evaluation would have been performed. Based on these (representative) data the authors could somehow extrapolate for the other study areas. However, such an approach must be thoroughly accompanied by statistical processes.
I wonder why the authors did not approach reknown allergology experts in the field in Japan to discuss the issues and how to overcome these points.
As recently outlined by Hossny et al in the World Allergy Organization Journal (WAO 2019) “a global increase in prevalence of asthma, allergic rhinitis, and atopic dermatitis, followed by a rapid rise in FA has been termed the "second wave of the allergy epidemic". Although we know of the global trend, the patterns of FA are highly variable in different parts of the world. These differences can be attributed partly to the many difficulties encountered in reaching an accurate estimation of prevalence. Self-report leads to an overestimating of prevalence by threefold to fourfold. Furthermore, widely varying study methodologies, the lack in the use of objective methods, and differences in baseline populations limit robust comparisons between populations.”
In particular in Asia data on food allergy prevalence is quite variable (Hossny et al, WAO 2019), and thus only proper methods will allow to define correct prevalence values.
The cumulative food allergy prevalence estimates of 10.3% and 12.2% at 1.5 and 3 years of age might be highly overestimated (as compared to data form US, Europe) or possibly in line with increasing prevalences of food allergy in Asia (as shown for Korea). However, without correct validation of the data we will not be able to substantiate this enigma.
